# Botryllin, a Novel Antimicrobial Peptide from the Colonial Ascidian *Botryllus schlosseri*

**DOI:** 10.3390/md21020074

**Published:** 2023-01-21

**Authors:** Nicola Franchi, Loriano Ballarin, Francesca Cima

**Affiliations:** 1Department of Life Science, University of Modena and Reggio Emilia, 41125 Modena, Italy; 2Department of Biology, University of Padova, 35131 Padova, Italy

**Keywords:** antimicrobial peptides, ascidians, *Botryllus*, haemocytes, tunicates

## Abstract

By mining the transcriptome of the colonial ascidian *Botryllus schlosseri*, we identified a transcript for a novel styelin-like antimicrobial peptide, which we named botryllin. The gene is constitutively transcribed by circulating cytotoxic morula cells (MCs) as a pre-propeptide that is then cleaved to mature peptide. The synthetic peptide, obtained from in silico translation of the transcript, shows robust killing activity of bacterial and unicellular yeast cells, causing breakages of both the plasma membrane and the cell wall. Specific monoclonal antibodies were raised against the epitopes of the putative amino acid sequence of the propeptide and the mature peptide; in both cases, they label the MC granular content. Upon MC degranulation induced by the presence of nonself, the antibodies recognise the extracellular nets with entrapped bacteria nearby MC remains. The obtained results suggest that the botryllin gene carries the information for the synthesis of an AMP involved in the protection of *B. schlosseri* from invading foreign cells.

## 1. Introduction

The increasing and indiscriminate use of antibiotics in industrialised countries has led to such a wide diffusion of resistant pathogens [1], the World Health Organisation (WHO) stated that the problem cannot be neglected anymore, and coordinated and global efforts are required to overcome it [2]. Within this context, a promising approach can derive from the study of the antimicrobial peptides (AMPs), components of innate immunity, widely diffuse in invertebrates and vertebrates, showing a high degree of convergent evolution in terms of primary, secondary, and tertiary structure and biological roles [3]. They are a class of small proteins, a few dozen amino acids long and less than 10 kDa in molecular weight, containing structurally segregated hydrophilic (mostly cationic) and hydrophobic residues. The latter enable the interaction with microbial plasma membrane and alter the integrity of the phospholipid bilayer [4,5,6,7,8,9]. Therefore, they induce, directly or indirectly, the lysis of the target cells [6,9,10,11]; in addition, some of them permeabilise the plasma membrane and allow the entry of other AMPs with intracellular targets [12]. This assures an effective protection against microorganisms associated with weak or no microbial resistance [13,14]. AMPs can be broadly divided into linear (further subdivided into alpha-helix, all beta, beta hairpin, and non-regular) and cyclical peptides [3]. They play a fundamental role in the innate immune responses of multicellular organisms, exerting antiviral and antifungal activity, in addition to their antibacterial properties [9,11,15]. Furthermore, AMPs can modulate the activity of the immune system [16], hamper cancer development [17], shape the gut microbiota [18], and exert control on bacterial endosymbionts [19,20,21,22].

Marine invertebrates represent the largest biodiversity of multicellular eukaryotes. They are a rich source of natural products [23,24,25], including AMPs [25,26]. Tunicates (phylum Chordata) are considered the sister group of vertebrates [27]. Ascidians are the most abundant tunicate group, with 2300 of the 3000 known species included in this taxon. Many families of cytotoxic compounds from tunicates have been described [28]. They include alkaloids, polyketides, cyanobactines (cyclic peptides produced by symbiotic bacteria), and AMPs [25,29,30]. The last are mainly produced by circulating haemocytes. Styelins A and E of *Styela clava* contain 32 amino acids with an abundance of phenylalanine and various modified residues [31]. They are synthesised by cytotoxic granulocytes [31] and show similarities with the cecropins, described in Diptera and Lepidoptera [32]. Clavanins A-E and clavaspirin are histidine-rich, 23 amino-acid-long peptides also synthesised and released by circulating cytotoxic cells of *S. clava* [33,34]. The octapeptideplicatamide is an AMP synthesised by the haemocytes of the congeneric species *Styela plicata* [35]. Halocyamines A and B of *Halocynthia roretzi* are tetrapeptides produced by circulating cytotoxic cells [36]; their diphenol rings represent suitable substrates for the enzyme phenoloxidase (PO), which explains their cytotoxic activity. Indeed, PO, by oxidizing the polyphenol substrata, produces reactive oxygen species and induces oxidative stress [37]. In *Ciona intestinalis* and *Ciona robusta*, two types of α-helix-AMPs are synthesised by cytotoxic haemocytes; the corresponding genes enhance their transcription upon the injection of nonself material in the body wall [38,39,40,41]. In some cases, AMPs are produced by symbiotic bacteria. For instance, didemnins are cyclic depsipeptides with antimicrobial activity isolated from species of the genus *Trididemnum*, with a molecular weight ranging from 0.94 to 1.1 kDa, and synthesised by endosymbiotic α-proteobacteria of the genus *Tistrella* [42,43,44].

*Botryllus schlosseri* is a colonial ascidian living in shallow waters of all the seas and oceans. It represents a widely used model for studies of tunicate immunobiology [45]. As an invertebrate, it relies only on innate immunity for its defence, and circulating immunocytes are the main effectors of immune responses. The latter include phagocytes and cytotoxic morula cells (MCs) [46]. Phagocytes exert their immunosurveillance activity by ingesting foreign material having entered the circulation, whereas MCs induce inflammation and cytotoxicity upon the recognition of nonself molecules [45]. MCs are also the main source of various complement components [47,48,49] and influence the activity of phagocytes [45,50,51]. In this work, we report on the identification, by mining the *B. schlosseri* transcriptome, of a transcript for a putative styelin-like AMP that we named botryllin, which is actively transcribed by MCs. The synthetic peptide, obtained from in silico translation of the transcript, exerts a toxic activity towards bacterial and unicellular yeast cells. Specific antibodies raised against epitopes of the putative amino acid sequence of the propeptide and the mature peptide label the MC granular content. The same antibody, upon MC degranulation induced by the presence of foreign cells, recognises the extracellular traps nearby MC remains. Collectively, the obtained results suggest that the botryllin gene carries the information for the synthesis of a novel AMP involved in the protection of *B. schlosseri* from invading foreign cells.

## 2. Results

### 2.1. Botryllin Identification

By BLAST analysis of our B. schlosseri EST collection and 3′ RACE, we identified a transcript for a protein with similarity (33.3% identity) to *S. clava* styelins D and E, based on phylogenetic analysis. The protein was named botryllin, and the sequence of the transcript was deposited in Genbank under the accession number OP851480 and was confirmed by alignment with the *B. schlosseri* genome. The gene organisation and the transcript sequences are reported in Appendix A. It codifies for a pre-propeptide of 158 amino acids, with a putative molecular weight of 18.11 kDa and a signal peptide of 23 residues. Analogously to styelins A–E from *S. clava* [32], the botryllin propeptide is likely split to the mature peptide of 37 amino acids that closely resembles the *S. clava* styelins (identity values ranging from 30 to 37.5%; Figure 1a). The deduced molecular weight of the mature peptide is 4.35 kDa, with a net positive charge, at pH 7.0, of 11.2.

According to CFSSP and PSIPRED prediction, 56.8% of the mature peptide has an alpha-helix secondary structure with a well-supported confidence of prediction (amino acids 4–10 and 22–35), whereas the prediction of a third alpha-helix domain (amino acids 14–16; 5.4% of the mature peptide) is not well supported (Figure 1b). PEP-FOLD3 predicted the 3D structure reported in Figure 1c. AMPA analysis supports its belonging to AMPs, with a mean antimicrobial value of 0.193, close to the value of Drosophila cecropin B.

### 2.2. Transcription Rate of the Botryllin Gene

Quantitative RT PCR analysis indicated no significant changes in the transcription rate of the botryllin gene up to 4 h of exposure to bacterial lipopolysaccharide (LPS; serotype 055:B5 from *E. coli*), *Bacillus clausii* cells, or zymosan particles from yeast cell walls for immune stimulation (data not shown).

### 2.3. Botryllin Has an Antimicrobial/Antimycotic Activity

The values of the minimal inhibitory concentration (MIC) and minimal bactericidal concentration (MBC) for the four bacterial strains and yeast cells are reported in Table 1. They suggest a real antimicrobial and antimycotic activity of the synthetic peptide, which is able to interfere with microorganism proliferation at relatively low concentrations, with the lowest value of MIC (3.1 µg mL^−1^) for both the two Gram-positive species (*B. clausii* and *Staphylococcus epidermidis*). Only with the Gram-negative species *Escherichia coli*, the MIC value amounted to 50 µg mL^−1^, whereas it resulted to 6.3 µg mL^−1^ in the case of *Proteus mirabilis* and *Saccharomyces cerevisiae* (Table 1).

The peptide was also able to prevent bacterial and fungal growth when the microbial strains were re-suspended in Brain Heart Broth Infusion (BHI), indicating a potential biocidal activity. The highest potential bactericidal and fungicidal activities have been observed for *S. epidermidis* and *S. cerevisiae*, respectively, which showed the lowest value of MBC (12.5 µg mL^−1^), whereas the MBC resulted higher than 50 µg mL^−1^ in all the other cases (Table 1).

The antimicrobial activity of botryllin was also assayed at various physico-chemical conditions, such as temperature, pH, and ionic strength, to determine the optimum and the resistance/decrease capability of the peptide (Table 2). In these experiments, *S. epidermidis* was chosen as a target strain for its sensitivity, as deduced on the basis of both MIC and MBC values. The predicted melting temperature (Tm) of botryllin, i.e., the temperature value at which half of its amino acids become destructured, resulted higher than 65 °C, as the amino acid sequence has a Tm index of 3.44. In agreement with the prediction, the antimicrobial activity of the peptide was not influenced by heating up to 100 °C, with a MIC value of 12.5 µg mL^−1^ after heating for 2 h at 60 °C and 80 °C, and of 6.3 µg mL^−1^ after heating for 15 min at 100 °C. The peptide is more active at pH 7.0, 8.0, and 9.0 (MIC value: 12.5 µg mL^−1^) with respect to lower pH. At pH 6.0, the MIC value was 50 µg mL^−1^, whereas at pH 5.0, the concentration required was higher than 50 µg mL^−1^. No MIC was determined at pH 4.0, as bacteria were not able to grow in this acidic condition (Table 2). As for the effects of the increasing ionic strength in the medium, the presence of 150 mM NaCl did not affect the antimicrobial activity (MIC value: 12.5 µg mL^−1^), whereas it was progressively decreased by increasing salt concentrations (25 and 50 µg mL^−1^ at 300 and 600 mM NaCl, respectively).

### 2.4. Botryllin Increases the Mortality of Bacterial and Yeast Cells

After 20 h of incubation in the presence of 6.25 µg mL^−1^ synthetic botryllin, 3 of the 4 bacterial strains (*B. clausii*, *E. coli*, and *P. mirabilis*) showed a significant (*p* < 0.05) increase in cell mortality (more than 60%, around 50%, and less than 40%, respectively). No significant differences were observed in the case of *S. epidermidis*. At 25 µg mL^−1^, all the bacterial strains showed significant (*p* < 0.05) higher mortality than the controls: it amounted to more than 60% for *B. clausii* and *E. coli*, and to a value around 50% for *S. epidermidis* and *P. mirabilis* (Figure 2a–d,f). Yeast cells were also killed by synthetic botryllin, with a significant (*p* < 0.05) difference with respect to the controls at 25 µg mL^−1^ (around 30% of dead cells) (Figure 2e,g). The presence of anti-botryllin Ab2 antibody partially reduced the negative effect of botryllin (25 µg mL^−1^) in the case of *B. clausii* and *E. coli*, whereas it completely inhibited the effect in the case of *S. epidermidis* and *P. mirabilis* and yeast cells (Figure 2a–e).

### 2.5. Botryllin Causes Irreversible Morphological Alterations of Microbial and Yeast Cells

Control microbial cells show a tight adhesion of the plasma membrane to the cell wall and a dense cytoplasm, with some paler regions corresponding to the nucleoid; some cells are dividing (Figure 3a,c and Figure 4a,e). The exposure to synthetic botryllin led to evident alterations to the cell morphology. At the concentration of 6.3 µg mL^−1^, the peptide induced, in *B. clausii*, the detachment of the cell wall and the disassembly of the peptidoglycans that appear as a series of filaments protruding from the remains of the former. The plasma membrane appeared damaged with the consequent release of cytoplasmic material (Figure 3b). At 6.3 µg mL^−1^, *S. epidermidis* cells showed the presence of numerous invaginations of the plasma membrane (Figure 3d), whereas at 25 µg mL^−1^, cell lysis was observed with disruption of the cell wall and the release of the cellular content (Figure 3e). At 6.3 µg mL^−1^, *E. coli* cells underwent a general shrinkage of the cytoplasm with the increase of the periplasmic space, between the plasma membrane and the cell wall, and the condensation of the DNA, which appeared adherent to the plasma membrane (Figure 4b). At 25 µg mL^−1^, the periplasmic space was generally increased, and many cells showed a damaged wall and the release of the cell content (Figure 4c,d). In *P. mirabilis*, a clear effect was observed at the concentration of the peptide of 25 µg mL^−1^: DNA condensed along the plasma membrane, the periplasmic space increased, and the cell wall lost its integrity (Figure 4f).

In the case of *S. cerevisiae*, control cells showed a normal budding activity (Figure 5a), which was not reported at 6.3 µg mL^−1^ of peptide. At this concentration, many vesicles appeared in the cytoplasm, which also resulted in less electron density than in controls; in addition, mitochondria showed an extensive altered morphology of cristae and an electron dense matrix (Figure 5b). At the peptide concentration of 25 µg mL^−1^, the cytoplasm shrunk, and the plasma membrane detached from the cell wall, a giant vacuole appeared in the cytoplasm, mitochondria disappeared, and the nucleus fragmented in a series of electron dense spots of condensed chromatin (Figure 5c).

### 2.6. Botryllin Is Synthesised by MC, Stored Inside Their Granules and Released upon MC Degranulation

In situ hybridization (ISH) analysis indicates that MCs are the only cells transcribing the *botryllin* gene (Figure 6a,b). The monoclonal anti-botryllin antibodies Ab1, Ab2, and Ab3 stained the granular content of MCs, indicating that the protein is stored inside MC granules (Figure 6c). The antibodies stained no other haemocyte type or tissue. This is further supported by observations under the TEM showing labelling inside the granules of MCs and of granular amoebocytes, the latter considered the MC precursors [46] (Figure 7a–d). Upon the exposure to microbial cells, MCs change their morphology and undergo degranulation: they release in the environment material that is recognised by the three antibodies and contributes to the formation of extracellular traps (ETs) found around MCs (Figure 6d,e). Electron microscopic analysis clearly indicates that immunopositivity to antibodies against the mature peptide (Ab1) is located in the fibrillary material released by MCs (Figure 7e), clearly indicating that active botryllin is a constituent of the extracellular traps formed by degranulating MCs.

### 2.7. B. clausii-Conditioned Medium (CM) Has Biocidal Activity That Is Inhibited by Anti-Botryllin Antibodies

*B. clausii*-derived CM significantly (*p* < 0.05) enhanced the fraction of dead *B. clausii* cells with respect to control. Analogously, the addition of synthetic botryllin at the concentration of 25 µg mL^−1^ led to a significant (*p* < 0.05) decrease of live cells. This increase was prevented by the addition of the anti-botryllin Ab2 or Ab3 antibodies, but not by preimmune mouse serum (Figure 8).

## 3. Discussion

AMPs represent a widespread component of innate immunity and an ancient defence strategy assuring the survival of organisms by preventing the entry and the spreading of potentially pathogenic microorganisms. Since they interact directly with the plasma membrane of the foreign cells, the development of a resistance by microorganisms is quite rare [9,11,15,52]. This led to an increasing interest in this class of molecules for their possible application in the biomedical field, which is currently hampered by their potential cytotoxicity, the high costs of production, and limited information on their bacteriostatic and bactericidal activity [3,53].

The colonial ascidian *B. schlosseri* is a reliable organism for the study of tunicate innate immunity, and various humoral factors involved in the protection from foreign cells have been identified so far, including the enzyme PO [54,55,56], a rhamnose-binding lectin with opsonic properties [57,58] and various complement components modulating the immune responses [47,48,49,59].

Two main immunocyte types are present in the *B. schlosseri* haemolymph: phagocytes and MCs [46,60]. The latter are the most abundant circulating haemocytes [46] and the first circulating cells sensing the presence of nonself [61]. They are large cells (≥10 µm in diameter), with the cytoplasm filled with many granules, uniform in size (around 2 µm in diameter) and containing the enzyme PO, responsible for the cytotoxic activity characterising these cells, and its polyphenol substrata [46]. As a consequence of the nonself recognition, MCs release cytokines, recruiting other immunocytes at the infection site, and undergo degranulation, with the release of their granular content [45]. The enzyme PO, acting on the polyphenol substrata, produces reactive oxygen species that induce a condition of oxidative stress in the nearby cells, responsible for the cytotoxic activity of MCs [55,62,63,64]. Among the released granular material, the protein p102 forms a network of amyloid fibres that contribute to the formation of extracellular traps able to prevent the spreading of microbes inside the colonial circulation [65].

In the present work, we identified, in the *B. schlosseri* transcriptome, a transcript for a peptide showing similarity with styelins D and E from *S. clava* [31,32,66], a solitary ascidian belonging to the clade Styelidae that includes also the genus *Botryllus*. Investigating the available transcriptomes and genome of *B. schlosseri*, we identified three similar sequences from only one transcriptome. They differ in a few nucleotides, both at the coding sequences and untranslated regions. The high sequence variability found in *B. schlosseri* was already noticed (e.g., [57]). This, together with a still highly fragmented genome, does not allow us to ascertain whether the above sequences are the result of a recent gene duplication or, more likely, are attributable to isoforms and not to different paralogous genes. Future studies will be aimed at the resolution of this problem.

The primary sequence of the mature peptide was obtained by in silico translation of the identified transcript, and the obtained amino acid sequence was compared with those of styelins. Our peptide, which we named botryllin, and styelins share a high percentage of identity (higher than 30%) and the full conservation of some amino acids and the KHK motif. The synthetic peptide was constructed on the mature peptide sequence and used for functional analyses.

We studied the MIC and the MBC of the synthetic peptide towards two Gram-positive and two Gram-negative bacteria species. In agreement with the computer prediction, botryllin shows in vitro antimicrobial activity against the target bacterial cells, as well as towards baker’s yeast cells. TEM analysis clearly indicates that synthetic botryllin induces severe morphological alterations in the tested microorganisms, ultimately leading to their death. However, it is conceivable that botryllin, analogously to styelin [67], has a broader spectrum of antimicrobial activity against a wide range of marine microorganisms that were not tested in the present study as they cannot be reared under laboratory conditions. In addition, similarly to styelins [66], various amino acids can be post-translationally modified so to further enhance the antimicrobial activity of the peptide. To verify this point, the biochemical purification of botryllin from the haemolymph of large colonies of *B. schlosseri* is required, and this will be a goal of our future research.

To evaluate the possible use of botryllin in a biotechnological or pharmaceutical context, we considered the resistance to denaturation at different temperatures and pHs. Thermal and pH stability are important, for instance, in treatments to get safe food without altering its taste, nutritional value, or colour [68,69]. Botryllin appears to be a stable molecule at different pHs and in a wide range of temperatures. This suggests a possible use of botryllin, once synthesised in large amounts in the laboratory, as a novel antibacterial compound in the pharmaceutical, medical, or veterinary fields, with a possible use in acid environments, such as the gastric lumen, where other AMPs can be denatured [67]. Its salt tolerance, as resulted from the experiments carried out at different ionic strengths, suggests its possible use as a new component of antifouling paints to be used in seawater without risks for the marine environment. Furthermore, the tolerance of botryllin to elevated salt concentrations can be exploited in those human pathologies, such as cystic fibrosis, characterised by high salt concentrations in the liquids moistening the lung cavities. The latter reduce defence responses and enhance the colonization of salt-resistant bacteria, such as *Pseudomonas aeruginosa* and *Staphylococcus aureus*, which can represent a serious risk for the life of patients [70]. However, studies on the cytotoxicity of botryllin towards mammalian cells and tissues are lacking, and this will be another aim of our future research.

Results obtained with the CM indicate that molecules with biocidal activity are released by haemocytes upon the recognition of microbes or, likely, also soluble nonself molecules. The significant decrease of the CM cytotoxic activity in the presence of anti-botryllin antibodies suggests that botryllin is present in the material(s) released by haemocytes.

ISH clearly indicates that botryllin is synthesised by MCs, the only immunocytes type actively transcribing the botryllin gene. The peptide is initially synthesised as a pre-propeptide of 158 amino acids, provided with a signal peptide of 23 residues, whereas the mature peptide results to 37 amino acids long, with a 3D structure including two short alpha helices. Immunocytochemical analysis indicates that it is stored inside MC granules; as stated above, upon the exposure to nonself, MCs degranulate and release their granular content, botryllin included. The absence of significant changes in the transcription level upon haemocyte exposure to nonself suggests that the increase in the molecule concentration in the infection sites is obtained by its release from the storage sites (MC granules), rather than through an increase in gene transcription. In addition, the labelling of MC granules by the Ab3 antibody, recognizing a propeptide epitope not included in the mature peptide, suggests that MC granules host the propeptide. Since the Ab3 antibody recognizes also the extracellular traps, we can suppose that its conversion to the mature peptide occurs in the extracellular environment, once the MC granular content has been released, through enzymatic proteolysis. In agreement with this hypothesis, we recently demonstrated the presence of the peptidase furin inside MC granules [65].

MC degranulation is usually associated with cytotoxicity and local cell death due to the contemporary release of the enzyme PO that, upon the oxidation of polyphenol substrata, produces reactive oxygen species (ROS; [63]). Interestingly, styelins (and probably botryllin) are DOPA-containing polypeptides [66] that can represent natural substrates of PO. This suggests that botryllin could exert its antimicrobial activity through both the alteration of the microbial plasma membrane, as reported for the majority of AMPs, and the production of ROS when oxidized by PO. Once released, botryllin is entangled in the net of amyloid material derived from the MC granular content [65] and becomes a constituent of the extracellular traps in order to increase the local concentration of AMP that helps in facing foreign cells and preventing their spreading in the circulation.

MCs are the effector cells of the allorejection reaction between contacting incompatible colonies [62,64]. As a consequence of the recognition of some still unknown content of the haemolymph plasma diffusing from the alien colony through the partially-fused tunics, they move towards the ampullae (the peripheral, sausage-like, blind endings of the circulation), facing the alien colony. Then, they migrate into the tunic and degranulate, contributing to the formation of the melanic, necrotic spots characterising the reaction [45,64]. It is conceivable that, as in the case of the recognition of microbes, botryllin is released also during the allorejection-associated degranulation, acting as a possible substrate for PO. This aspect, not yet investigated, suggests a possible role of the peptide in the protection from microbial invasion in the course of the allorejection reaction, during which the epithelium of the facing ampullar tips appear highly altered with the appearance of large fenestrations [71]. Future efforts will, therefore, also be aimed at investigating the possible role of botryllin in the allorejection reaction of *B. schlosseri*.

## 4. Materials and Methods

### 4.1. Animals

Colonies of *B. schlosseri*, collected near the marine station of the Department of Biology, University of Padova, in Chioggia (Southern part of the Lagoon of Venice), were attached to glass slides, transferred in aerated aquaria at a constant temperature of 19 °C and 12 h:12 h of light:dark. They were fed with unicellular algae (*Dunaliella* sp., *Tetraselmis* sp.).

### 4.2. Haemocyte Collection

The peripheral vessel of colonies, previously blotted dry, was punctured with a fine tungsten needle; flowing haemolymph was collected with a glass micropipette and transferred in a 1.5-mL vial. It was spun at 780× *g* at 4 °C for 10 min, and the supernatant was discarded. Pelleted haemocytes were re-suspended in filtered seawater (FSW) to the final concentration of 5 × 10^6^ cells mL^−1^.

### 4.3. Primer Design, RNA Extraction, cDNA Synthesis, Cloning, and Sequencing

Total RNA was isolated from colonies of *B. schlosseri* and haemocytes with the RNA NucleoSpin RNA XS (Macherey–Nagel, Düren, Germany) kit, and its quality was determined by the A_260/280_ ratio and visualised in Midori green (Nippon GeneticsEurope Gmbh, Düren, Germany)-stained 1.5% agarose gels. The first strand of cDNA was reverse transcribed from 1 μg of total RNA at 42 °C for 1 h in a 20 μL reaction mixture containing 1 μL of ImPromII Reverse Transcriptase (Promega, Madison, WI, USA) and 0.5 μg oligo(dT)-Anchor primer or random primers (Promega, Madison, WI, USA).

The primers reported in Table 3 were used for PCR reactions in a 25 μL reaction volume containing 1 μL of cDNA from *B. schlosseri* colonies, 2.5 μL of 10× incubation buffer (PCRBIO Classic Taq, PCR BIOSYSTEMS, London, UK) with 15 mM MgCl_2_, 0.25 μM of each primer, 10 mM of each of the deoxynucleotide triphosphates, and 2 units of Taq polymerase. PCR was performed on a MyCycler (BioRad; Hercules, CA, USA) thermocycler with the following conditions: 94 °C for 2 min, then 40 cycles of 94 °C for 30 s, 55–60 °C for 30 s, 72 °C for 60 s, and 72 °C for 10 min. Amplicons were separated by electrophoresis on 1.5% agarose gel, and the corresponding bands were purified with the ULTRAPrep Agarose Gel Extraction Mini Prep kit (AHN Biotechnologie, Nordhausen, Germany), ligated in pGEM-T Easy Vector (Promega, Madison, WI, USA), and cloned in DH-5α *E. coli* cells. Positively screened clones were Sanger sequenced at Eurofins Genomics (Ebersberg, Germany) on an ABI 3730XL Applied Biosystems apparatus (Life Technologies Europe BV, Monza, Italy). The 3′-rapid amplification of the cDNA ends (RACE) was performed using the 2nd Generation of the 5′/3′RACE Kit (Roche, Basel, Switzerland). In order to obtain the 3′ sequence of *botryllin* cDNA, the same forward primers used for RT-PCR and probe synthesis (Table 3) were used for nested PCR with anchor reverse primer, according to the manufacturer’s instructions.

### 4.4. Three-Dimensional Structure of Botryllin

After in silico translation of the identified transcript with Expasy (https://web.expasy.org/translate/ (accessed on 23 October 2022)), the secondary structure of the peptide was predicted using CFSSP (http://www.biogem.org/tool/chou-fasman (accessed on 25 October 2022)) and PSIPRED (http://bioinf.cs.ucl.ac.uk/psipred (accessed on 25 October 2022)). The sequence was submitted to PEP-FOLD3 (https://bioserv.rpbs.univ-paris-diderot.fr/services/PEP-FOLD3 (accessed on 23 October 2022)) for the prediction of the 3D structure. AMPA (antimicrobial sequence scanning system; http://tcoffee.crg.cat/apps/ampa/do (accessed on 4 November 2022)) was used to evaluate the possible antimicrobial activity of the identified peptide.

### 4.5. Quantitative Real-Time PCT (qRT-PCR)

After the exposure to LPS (Sigma Aldrich, St. Louis, MO, USA), *B. clausii* or zymosan (Sigma Aldrich, St. Louis, MO, USA), haemocytes were collected, and their mRNA was extracted as reported before. Relative qRT-PCR was carried out, according to the method reported in Franchi and Ballarin [47], to estimate the relative variation of mRNA for botryllin. In this case, unexposed cells were considered as the reference control. Forward and reverse specific primers for the above-reported transcripts and for the elongation factor 1α (EF1α) were designed and reported in Table 3. All the designed primers contained parts of contiguous exons to exclude contamination by genomic DNA; a qualitative PCR was also carried out before qRT-PCR. In addition, the analysis of the qRT-PCR dissociation curve gave no indications of the presence of contaminating DNA. The following cycling parameters were used: 10 min at 95 °C (denaturation), 15 s at 95 °C plus 1 min at 60 °C, 15 s at 95 °C for 45 times, 1 min at 60 °C. Each set of samples was run three times on an Applied Biosystem 7900 HT Fast Real-Time PCR System (Life Technologies Europe BV, Monza, Italy), and each plate contained cDNA from three different biological and control samples (*n* = 3). The 2^−∆∆CT^ method [72] was used to estimate the relative amount of mRNA. The amounts of transcripts under different conditions were normalised to EF1α to compensate for variations in the amounts of cDNA.

### 4.6. In Situ Hybridization (ISH)

Using the primers reported in Table 3, we produced the DIG-labelled antisense riboprobes, as previously described [48]. Collected haemocytes were left to adhere for 30 min on SuperFrost Plus (Menzel–Glaser, Braunschweig, Germany) glass slides. Cells were then incubated for 1 h in FSW in the presence or in the absence (control) of either LPS (1 mg mL^−1^; serotype 055:B5 from *E. coli*), *B. clausii* (4 × 10^8^ cells mL^−1^), or zymosan (1 mg mL^−1^). After washing in FSW, they were fixed for 30 min at 4 °C in 4% paraformaldehyde plus 0.1% glutaraldehyde in 0.4 M cacodylate buffer, containing 1.7% NaCl and 1% sucrose. They were then permeabilised in a solution of 0.1% Triton X in phosphate-buffered saline (PBS: 1.37 M NaCl, 0.03 M KCl, 0.015 M KH_2_PO_4_, 0.065 M Na_2_HPO_4_, pH 7.2) for 5 min, washed in PBS and pre-incubated in Hybridization Cocktail 50% formamide (Amresco, Solon, OH, USA) for 1 h at 60 °C, and hybridised in the same solution containing 1 μg mL^−1^ riboprobe, overnight, at the same temperature. Cells were then washed in saline sodium citrate (SSC: 0.3 M NaCl, 40 mM sodium citrate, pH 4.5), for 5 min, and in a solution of 50% formamide in SSC at 60 °C, for 30 min, followed by an additional washing in PBS containing 0.1% Tween 20 (PBST) at room temperature, for 5 min. Haemocyte monolayers were then incubated in 1% powdered milk in PBST for 1 h (to reduce unspecific staining), followed by 5% methanol for 30 min (to block endogenous peroxidases). Cells were finally incubated at 4 °C overnight in an anti-DIG-monoclonal antibody labelled with alkaline phosphatase (Fab fragment; Roche Diagnostics, Basel, Switzerland), diluted 1/3000 in PBST. Samples were then stained with a solution of 0.3% nitro-blue tetrazolium/5-bromo-4-chloro-3-indolyphosphate- p-toluidine (NBT/BCIP) in Na-Tris-Mg buffer (NTM: 0.05%MgCl_2_ 1M, 0.05% NaCl 2M, 0.1% Tris-HCl 2M, pH 9.5) for 10 min and washed in PBS. Slides were then washed in distilled water and mounted in Eukitt (Electron Microscopy Sciences, Hatfield, PA, USA) before their observation under the light microscope. DIG-labelled sense probes were used as negative controls.

### 4.7. Synthetic Peptide and Specific Antibodies

The synthetic peptide and the antibodies were produced by Abmart Inc. (Shanghai, China). The synthetic peptide fully respected the sequence of 37 amino acids of the mature peptide. Three monoclonal antibodies were obtained by the same company by immunising BALB/c mice with the sequences of three highly immunogenic peptides from the pro-peptide sequence (Table 4). Pre-immune mouse serum was obtained by collecting the supernatant of freshly collected mouse blood after its centrifugation for 10 min at 780× *g* at 4 °C.

### 4.8. Immunocytochemistry

Exposed haemocytes were fixed as described above, washed in PBS, permeabilised in PBST, and incubated for 30 min to 1% Evans Blue (to quench autofluorescence). Cells were then incubated for 30 min in 10% goat serum in PBS (to reduce nonspecific staining) and with Ab1, Ab2, or Ab3, at the concentration of 10 µg mL^−1^, for 1 h. After washing in PBS, haemocytes were incubated for 1 h in biotin-conjugated goat anti-mouse IgG antibody (Calbiochem, San Diego, CA, USA), 10 µg mL^−1^ in PBS, washed again, and exposed to FITC-conjugated streptavidin (Sigma Aldrich, St. Louis, MO, USA). After a final washing, slides were mounted with FluorSave^TM^ Reagent (Calbiochem, San Diego, CA, USA) and observed under an Olympus (Tokyo, Japan) CX31 light microscope (LM) equipped with an LED fluorescence module with an exciting wavelength of 450 nm (Amplified by Fluorescence Excitation of Radiation Transmitted, AFTER, Fraen Corp. s.r.l., Milan, Italy).

### 4.9. Minimal Inhibitory Concentration (MIC) and Minimal Bactericidal Concentration (MBC) Determination

For the evaluation of MIC and MBC, four bacterial strains were used: *B. clausii* SIN, O/C, T, N/R, and *S. epidermidis* ATCC 700565 as Gram-positive and *E. coli* O4 and *P. mirabilis* ATCC 29906 as Gram-negative. With the exception of *B. clausii* (commercial *Enterogermina*, Sanofi, Opella Healthcare Italy S.r.l., Origgio, Italy),they were gently provided by Prof. G. Bertoloni, Department of Histology, Microbiology and Medical Biotechnologies, University of Padova, and Prof. R. Provvedi, Department of Biology, University of Padova. In addition, baker’s yeast (*S. cerevisiae*) was also used to evaluate the antimycotic activity of botryllin.

Bacteria were grown in sterile brain heart infusion (BHI; Honeywell-Fluka, Charlotte, NC, USA), and the microbial concentration was assessed by the determination of the absorbance at 580 nm with a Beckman Coulter Life Science (Brea, CA, USA) DU 730 UV/Vis spectrophotometer.

For MIC assessment, 50 µL of synthetic botryllin (100 µg mL^−1^ in BHI) were serially diluted in the wells of a 96-well, flat-bottomed microplate, and 50 µL of bacterial or yeast suspension, at the concentration corresponding to 0.01 absorbance units, were added to each well. Cells were incubated for 20 h at 37 °C, and the absorbance at 580 nm of the cell suspensions was then determined with a Plate Reader photometer (Elettrofor Scientific Instruments, Borsea, (RO), Italy) to evaluate the microbial growth.

For the evaluation of MBC, the content of the wells with the absence of growth was collected, centrifuged at 10,000× *g*, re-suspended in 100 µL of BHI, and transferred to new wells of a 96-well, flat-bottomed microplate. It was incubated for 20 h at 37 °C, and the absorbance of the wells at 580 nm was finally read with the microplate reader.

### 4.10. Microbial Cell Mortality Assay

The mortality of the bacterial strains was evaluated using the LIVE/DEAD BacLight Bacterial Viability kit for microscopy (Molecular Probes, Eugene, OR, USA). Briefly, 100 µL of bacterial cells at the concentration corresponding to 0.01 absorbance units were incubated for 20 h at 37 °C with 0 (control), 6.3, and 25 µg mL^−1^ of synthetic botryllin in BHI, and with 25 µg mL^−1^ of synthetic botryllin pre-incubated for 30 min with 10 µg mL^−1^ of anti-botryllinAb2 antibody. In another series of experiments, *B. clausii* cells were incubated in the presence or in the absence (control) of CM (see below) or CM and anti-botryllin Ab2 or Ab3 antibodies (pre-immune mouse serum in controls). Then, cells were pelleted by centrifugation and re-suspended in 0.85% NaCl. An equal volume of solutions A and B of the Bacterial Viability kit were mixed together and added to the bacterial suspensions in the ratio of 3:1 (vol:vol). After 15 min of incubation, cells were pelleted as described above and re-suspended in 50 µL of 0.85% NaCl, and 5 µL of the resulting suspension were laid on a Superfrost^TM^ Plus (Thermo Scientific, Waltham, MA, USA) glass slide and observed under the light microscope equipped with fluorescence module. Live cells appeared green when the excitation light of 535 nm was used, whereas dead cells resulted in red when excited with blue light (450 nm). Images were acquired with a Lumenera Infinity 2 camera, and the percentage of living and dead cells was determined by analysing images with the Infinity Analyze 5.0.0 software (Lumenera Corporation, Teledyne Technologies Inc., Thousand Oaks, CA, USA).

The effects of the synthetic peptide on yeast mortality were evaluated with the Trypan Blue exclusion assay [73]. Yeast cells, previously exposed to 6.3 and 25 µg mL^−1^ of synthetic botryllin in BHI, and with 25 µg mL^−1^ of synthetic botryllin pre-incubated with 10 µg mL^−1^ of anti-botryllin Ab2 antibody, were incubated in a 0.4% solution of Trypan Blue for 5 min and observed under the light microscope. Dead cells resulted as blue-stained.

### 4.11. Temperature, pH and Salinity Stability Assays

We used *S. epidermidis* for the evaluation of the effects of variations of physico-chemical parameters on the antibacterial activity of botryllin by evaluating the MIC values. The synthetic peptide, prepared as described above, was exposed at 60 and 80 °C for 4 h and at 100 °C for 15 min. It was then cooled at room temperature and used for MIC assay. In addition, the Tm of botryllin, i.e., the temperature value at which half of its amino acids result as destructured, was predicted with the software Tm Predictor (http://tm.life.nthu.edu.tw (accessed on 14 September 2022)).

To evaluate the effects of pH, the peptide was diluted in BHI prepared with 0.1 M Na-acetate buffer at pH 4.0 and 5.0; with 0.1 M phosphate buffer at pH 6.0, 7.0, and 8.0; and with 0.1 M Tris-HCl at pH 9.0.

The effects of the ionic strength on antimicrobial activity were assayed by MIC determination of the peptide diluted in BHI containing 150, 300, and 600 mM NaCl.

### 4.12. Electron Microscopy

Microbial and yeast cells, previously incubated with the synthetic peptide for 20 h at 37 °C, were pelleted by centrifugation; fixed in 2% glutaraldehyde in 0.2 M Na-cacodylate buffer, pH 7.2, for 2 h at 4 °C; dehydrated in an increasing ethanol series; and embedded in Epon (Fluka). Ultrathin sections (60–70 nm) were stained with uranyl acetate and lead citrate. They were finally observed under a FEI TECNAI 12 transmission electron microscope (TEM), at 75 kV, equipped with a TIETZ high-resolution digital camera.

For immunocytochemical analyses, samples were dehydrated in increasing ethanol concentrations and embedded in London Resin White (LRW, Polyscience, Warrington, PA, USA). Ultrathin sections, once collected on copper grids, were incubated for 10 min in 10% goat serum, washed in PBS, incubated overnight in the primary antibody (Ab1) at the concentration reported above, washed again, and treated for 1 h with goat anti-mouse IgG antibody (10 µg mL^−1^) conjugated with colloidal gold spherules (15 nm in diameter). Grids were then washed in PBS and in distilled water and stained with uranyl acetate and lead citrate. Sections were finally observed under the TEM, as reported above.

### 4.13. Preparation of Conditioned Medium (CM)

*B. schlosseri* haemocytes, collected as described above, were incubated for 30 min in FSW in the presence or in the absence (control) of 2 × 10^9^
*B. clausii* spores (*Enterogermina*, Sanofi, Opella Healthcare Italy S.r.l., Origgio, Italy). They were then centrifuged at 2000× *g* for 5 min, the supernatant was collected, and 1% (vol:vol) of Protease Inhibitor Mix (GE Healthcare Europe GmbH, Freiburg, Germany) was added before its storage at −20 °C until use as CM in microbial cell mortality assay, as described above.

### 4.14. Statistical Analysis

All the experiments were repeated with three different colonies (*n* = 3). Each experiment was carried out in triplicate. The fraction of labelled cells was compared with the chi square test, whereas qRT-PCT data were analysed with the one-way ANOVA followed by the Duncan’s test for multiple comparisons with DSAASTAT v. 1.1 2011 [74]. Differences were considered statistically significant when *p* < 0.05.

## Figures and Tables

**Figure 1 marinedrugs-21-00074-f001:**
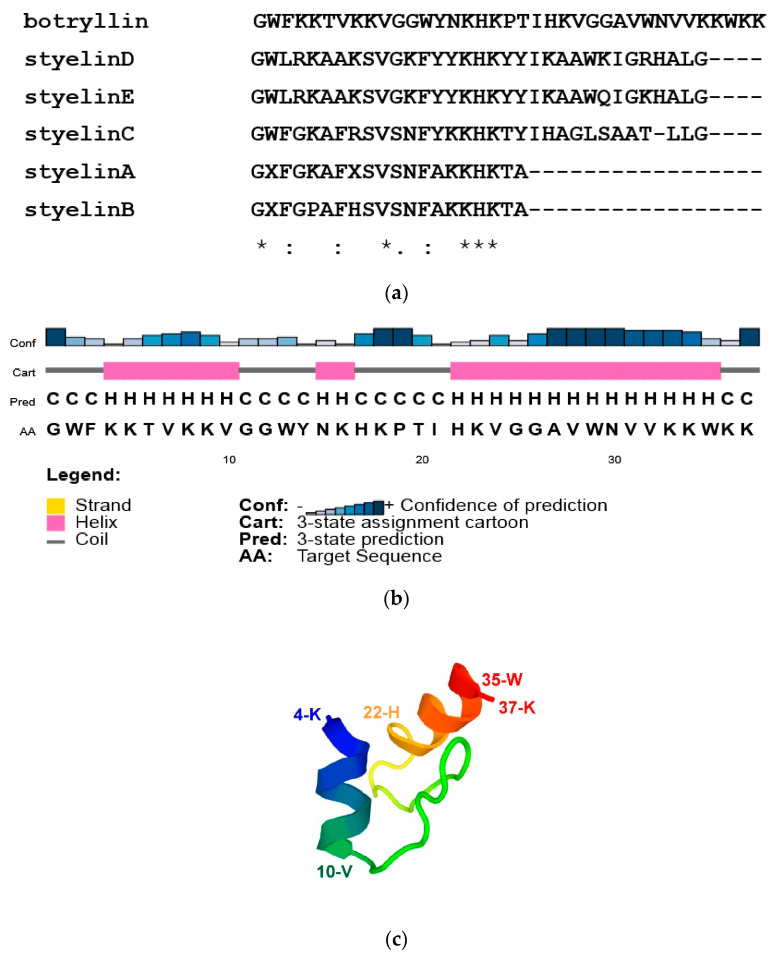
(**a**) Alignment of *B. schlosseri* botryllin and *S. clava* styelins. (**b**,**c**) Predicted structures of botryllin. (**b**) Secondary structure according to PSIPRED analysis. (**c**) 3D structure according to PEP-FOLD3.

**Figure 2 marinedrugs-21-00074-f002:**
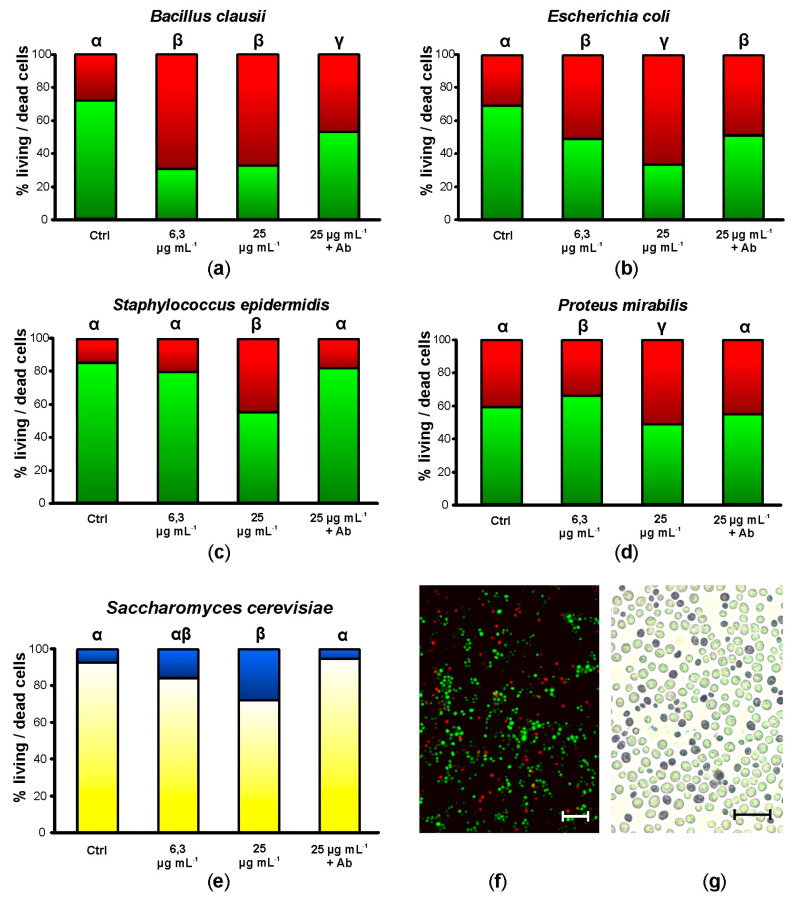
Mortality assay in the presence of botryllin. (**a**–**d**) Percentage of living (green) and dead (red) bacterial cells after 20 h of incubation in the presence of 6.3 and 25 µg mL^−1^ of synthetic botryllin, and 25 µg mL^−1^ of botryllin pre-incubated for 30 min with 10 µg mL^−1^ of anti-botryllin Ab2 antibody, as evaluated with the LIVE/DEAD BacLight Bacterial Viability kit; (**e**) percentage of living (yellow) and dead (blue) yeast cells after the treatments described above, evaluated with the Trypan Blue exclusion assay; (**f**) *S. epidermidis* cells labelled with the LIVE/DEAD after the exposure to 25 µg mL^−1^ of synthetic botryllin: dead cells appear red; (**g**) *S. cerevisiae* cells treated with Trypan Blue after the exposure to 25 µg mL^−1^ of synthetic botryllin: dead cells appear blue. Statistically significant (*p* < 0.05) differences in (**a**–**e**) are marked by different Greek letters on the top of the bars. Scale bars: 6 µm in (**f**), 30 µm in (**g**).

**Figure 3 marinedrugs-21-00074-f003:**
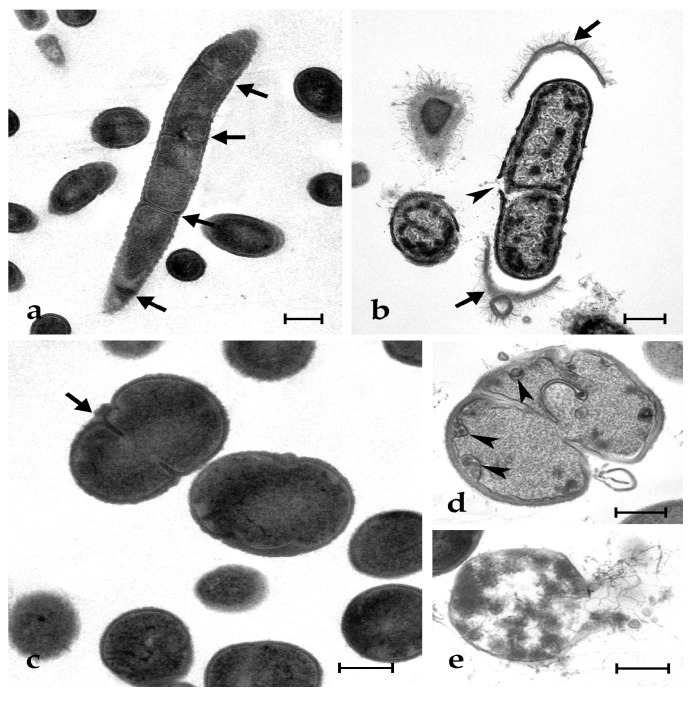
Transmission electron micrographs of *B. clausii* (**a**,**b**) and *S. epidermidis* (**c**–**e**) cells. (**a**,**c**) Control (untreated) cells: arrows indicate cleavage furrows; (**b**) *B. clausii* cells after exposure to 25 µg mL^−1^ of synthetic botryllin. Arrows show the detachment of the cell wall and the arrowhead indicates the breakage of the plasma membrane; (**d**,**e**) *S. epidermidis* cells after the exposure to botryllin at the concentration of 6.3 and 25 µg mL^−1^, respectively. Arrowheads in (**d**) indicate plasma membrane invaginations. Note in (**e**) the cell lysis after exposure to the highest peptide concentration. Scale bars: 0.5 µm.

**Figure 4 marinedrugs-21-00074-f004:**
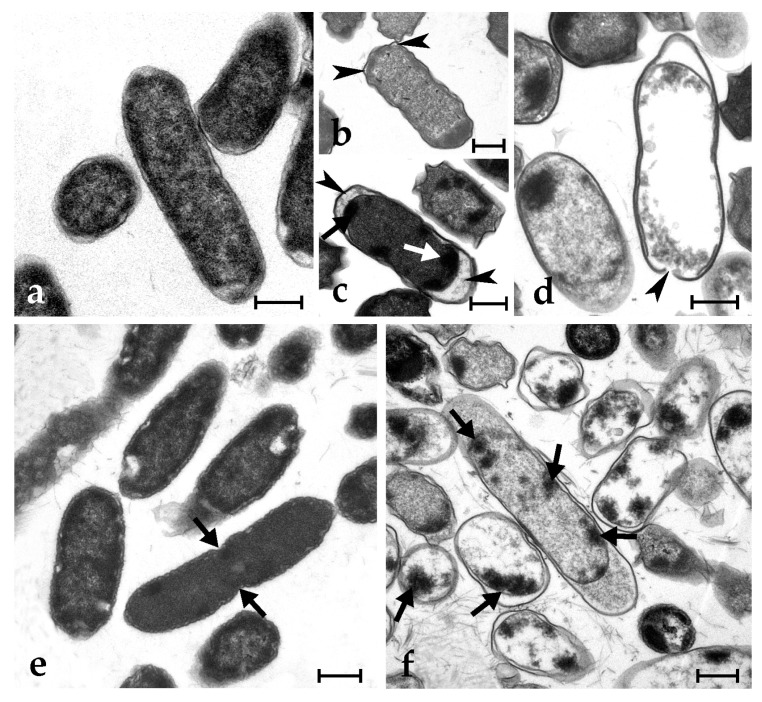
Transmission electron micrographs of *E. coli* (**a**–**d**) and *P. mirabilis* (**e**,**f**) cells. (**a**,**e**) Control (untreated) cells: arrows indicate a cleavage furrow; (**b**–**d**) *E. coli* cells after exposure to 6.3 (**b**) and 25 (**c**,**d**) µg mL^−1^ of synthetic botryllin. Arrowheads indicate the enlargement of the periplasmic space whereas arrow marks condensed DNA adherent to the plasma membrane; (**f**) *P. mirabilis* cells after the exposure to botryllin at the concentration of 25 µg mL^−1^. Arrows indicate unusual DNA condensation along the periphery of the cell. Scale bars: 0.5 µm.

**Figure 5 marinedrugs-21-00074-f005:**
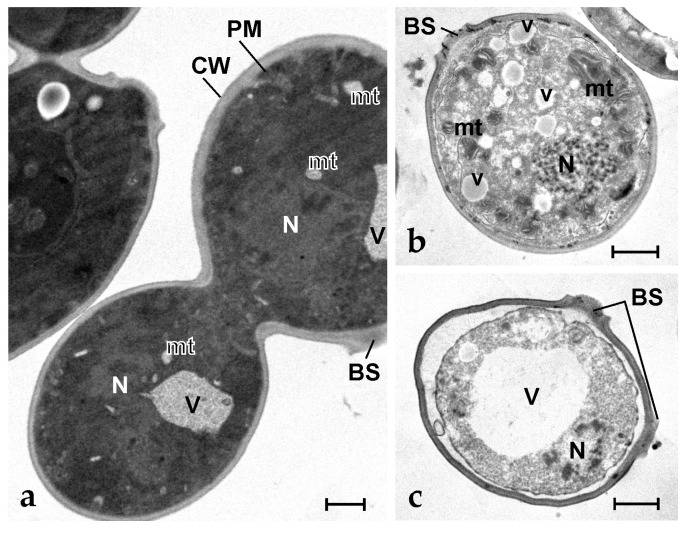
Transmission electron micrographs of *S. cerevisiae* cells. (**a**) Control (untreated) dividing cells; (**b**,**c**) Yeast cells after exposure to 6.3 µg mL^−1^ of synthetic botryllin. BS: budding scar; CW: cell wall; mt: mitochondrion; N: nucleus; V: vacuole; v: small vesicle. Scale bars: 1 µm.

**Figure 6 marinedrugs-21-00074-f006:**
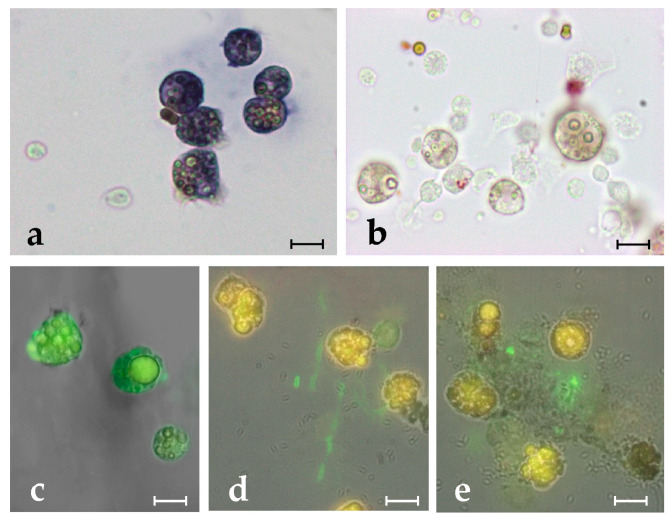
(**a**,**b**) ISH on *Botryllus* haemocytes with botryllin riboprobe. (**a**) Antisense probe: positive cells, represented by MCs, stain blue; (**b**) Sense probe: MCs appear unstained; (**c**–**e**) Immunofluorescence of haemocyte smears incubated with anti-botryllin Ab1 (**c**,**d**) and Ab3 (**e**) antibodies. (**c**) Immunopositive (green fluorescence) control MCs; (**d**,**e**) Immunopositive (green fluorescence) fibrillary material, forming extracellular traps, released in the presence of microbial cells (*S. epidermidis*), by MCs that appear unlabelled by the anti-botryllin antibody and only show the typical yellow autofluorescence of their vacuolar polyphenolic content. Scale bars: 10 µm.

**Figure 7 marinedrugs-21-00074-f007:**
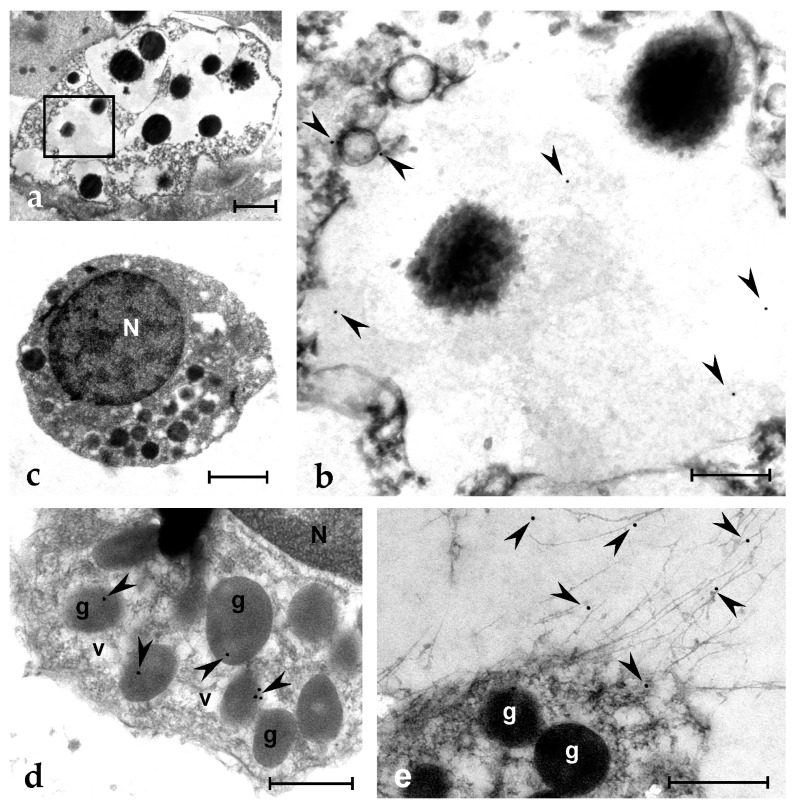
TEM micrographs of *Botryllus* haemocytes treated with the Ab1 anti-botryllin antibody; immunogold technique. (**a**) MC with large granules containing electron-dense material. An enlargement of the squared region is shown in (**b**), where some gold spherules (arrowheads) are visible. (**c**) A granular amoebocyte, precursor of MC, with the electron-dense granules; (**d**) Detail of a cytoplasmic region of a granular amoebocytes with labelled granules (arrowheads); (**e**) Immunopositive (arrowheads) extracellular fibrillary material released by MCs. g: granule; N: nucleus; v: small vesicle. Scale bars: 2 µm in (**a**), 0.5 µm in (**b**,**d**,**e**); 1 µm in (**c**).

**Figure 8 marinedrugs-21-00074-f008:**
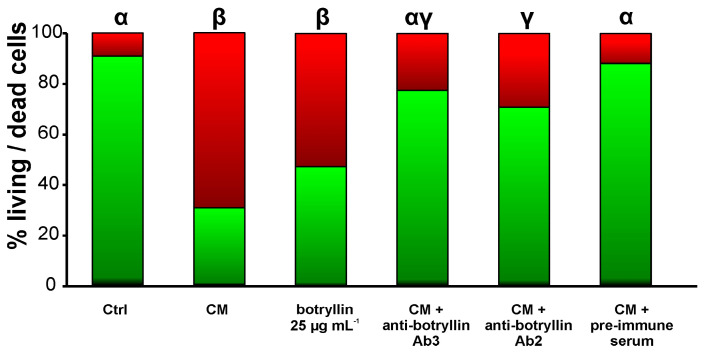
Percentage of living (green) and dead (red) *B. clausii* cells after 20 h of incubation in the presence of CM from *B. clausii*-exposed haemocytes, botryllin (25 µg mL^−1^), CM plus anti-botryllin Ab2 or Ab3 antibodies, CM plus pre-immune mouse serum. Cell mortality was evaluated with the LIVE/DEAD BacLight Bacterial Viability kit. Significant (*p* < 0.05) differences among the various conditions are expressed with different letters.

**Table 1 marinedrugs-21-00074-t001:** MIC and MBC values of botryllin on four bacterial strains and yeast cells.

Microorganism	MIC (µg mL^−1^)	MBC (µg mL^−1^)
*Bacillus clausii*	3.1	>50.0
*Staphylococcus epidermidis*	3.1	12.5
*Escherichia coli*	50.0	>50.0
*Proteus mirabilis*	6.3	>50.0
*Saccharomyces cerevisiae*	6.3	12.5

**Table 2 marinedrugs-21-00074-t002:** MIC values of botryllin towards *S. epidermidis* at various temperatures, pH values, and ionic strength.

Temperature (°C)			pH				Ionic Strength [NaCl] (mM)		
60 (2 h)	80 (2 h)	100 (15 min)	6.0	7.0	8.0	9.0	150	300	600
12.5	12.5	6.3	50.0	12.5	12.5	12.5	12.5	25.0	50.0

**Table 3 marinedrugs-21-00074-t003:** Primers used in the present work.

Primer	Sequence
Botryllin forward	CTGGTTTCTCCAATAACG
Botryllin reverse1	CAAGGTCATATTGGTGGCTA
Botryllin reverse2	GTCGAAGCTGTGCAAGACAT
Botryllin-qRT PCR For	GGTCGGGGGATGGTATAAT
Botryllin-qRT PCR Rev	ACCATCGTATTCGTCCCG
BsEF1-alpha qPCR For	GCCGCCATACTCTGAAGC
BsEF1-alpha qPCR Rev	GTCCAACTGGCACTGTTCC
Botryllin ISH For	GGTCGGGGGATGGTATAAT
Botryllin ISH Rev	CAAGGTCATATTGGTGGCTA

**Table 4 marinedrugs-21-00074-t004:** Peptides used for the production of monoclonal antibodies.

Epitopes	Sequence Location	Antibody
GWYNKHKPTIHK	N-terminus of the mature peptide	Ab1
VKGWGKDSDKEL	C-terminus of the mature peptide and beyond	Ab2
KEHDRDEYDGAL	part of the propeptide beyond the C-terminus of the mature peptide	Ab3

## Data Availability

All data generated or analysed during this study are included in this published article. The sequence of the botryllin transcript has been deposited in GenBank as described in Material and Methods.

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
