# Peer review of "Botryllin, a Novel Antimicrobial Peptide from the Colonial Ascidian Botryllus schlosseri"

_marinedrugs, 2023, doi:10.3390/md21020074_

Round 1

Reviewer 1 Report

In this manuscript, Franchi and colleagues report a novel styelin-like antimicrobial peptide named botryllin from the colonial ascidian B. schlosseri, characterizing the biological properties of the synthetic peptide and its activity towards a panel of bacterial strains and yeast. The authors accompanied routine MIC/MBC tests with straightforward  interpretation with nice microscopic photographs that demonstrate the significant morphological alterations occurring in target cells upon the exposure to the peptide. Moreover, they were able to locate the peptide to MC granules and to demonstrate its release in response to the exposure to microbial cells, which is certainly a relevant finding for the understanding of the role played by this AMP in the context of innate immune response.
Overall, the information presented is novel and signficance of the findings is relevant for the developmental and comparative immunology community.

L28: “with a high degree of evolutionary conservation”. One might argue that, with a few exceptions, AMPs rarely display a broad taxonomical distribution, being on the other hand often lineage-specific. I would suggest the authors here to emphasize the aspects of convergent evolution, both in terms of sequence (e.g. towards similar chemico-physical properties, or shared 2D and 3D structures) and in terms of biological role (e.g. acting as a first line of defense). In its present form, this sentence seems to suggest that most AMPs have a shared evolutionary origin.

L43: from an evolutionary point of view, this sentence is a somewhat confusing, considering that marine invertebrates are definitely not monophyletic (hence the use of the term “radiation” may seem inappropriate) and that arthropods are by far the largest animal phylum.

L48: if the authors here are referring to the mature peptide (and not the full-length precursor), then this should be mentioned.

L84 and elsewhere in the text: please check that all scientific names are in italics

L92: has the processing of the precursor peptide been previously characterized in other ascidians? I guess that, upon signal peptide cleavage, the precursor is expected to undergo further proteolytic cleavage to generate the 46aa-long mature peptide.

Figure 1: it would have been interesting to see here a multiple sequence alignment between botryllin and previously characterized styelins.

Figure 2: the use of letters to indicate both the figure panels and the statistical significance may be confusing. I would suggest to replace the latter with different symbols.

L211: since the genome of the target species is available, were the authors able to rule out the possibility that other paralogous genes were present? This might be relevant to check that ISH signals were specific to this particular transcript.

L284: while the use of a panel of microorganisms of common laboratory use is certainly useful to preliminarily investigate the antimicrobial activity of a novel AMP, I would suggest the authors to remark the fact that botryllin might have a broader spectrum of activity, perhaps showing even stronger killing activity, against a plethora of other microorganisms found in the marine environment that were not tested in the present study (and could not be tested due to the facts that the overwhelming majority of marine bacteria cannot be presently cultured in laboratory conditions).

L304: this is a reasonable consideration, even though the authors cannot exclude that other biological stimuli (e.g. the exposure to other microbes commonly found in the marine environment) can trigger botryllin expression. On the other hand, the massive depletion of botryllin from MC granules would suggest a regulation mode that could involve apparent over-expression several hours after the exposure, in order to replace the AMP storage lost with degranulation.

Author Response

We thank the anonymous reviewer 1 for his/her comments that helped us to improve the quality of the manuscript. Here below our replies to their comments

L28: “with a high degree of evolutionary conservation”. One might argue that, with a few exceptions, AMPs rarely display a broad taxonomical distribution, being on the other hand often lineage-specific. I would suggest the authors here to emphasize the aspects of convergent evolution, both in terms of sequence (e.g. towards similar chemico-physical properties, or shared 2D and 3D structures) and in terms of biological role (e.g. acting as a first line of defense). In its present form, this sentence seems to suggest that most AMPs have a shared evolutionary origin.

The sentence has been changed accordingly.

L43: from an evolutionary point of view, this sentence is a somewhat confusing, considering that marine invertebrates are definitely not monophyletic (hence the use of the term “radiation” may seem inappropriate) and that arthropods are by far the largest animal phylum.

We deleted the second part of the sentence.

L48: if the authors here are referring to the mature peptide (and not the full-length precursor), then this should be mentioned.

Done

L84 and elsewhere in the text: please check that all scientific names are in italics

Done

L92: has the processing of the precursor peptide been previously characterized in other ascidians? I guess that, upon signal peptide cleavage, the precursor is expected to undergo further proteolytic cleavage to generate the 46aa-long mature peptide.

The reviewer is right. Given the similarity of botryllin with styelins, there should be various proteolytic steps to get the mature peptide starting from a pre-propeptide, including the loss of the signal peptide and of the remaining parts of the propeptide not included in the mature peptide. Accordingly, the latter should be 37 aa-long. We, therefore, modified our previous assumptions in the text. It is our intention to conform this assumption by purifying, in the next future, the botryllin from the haemolymph of Botryllus.

Figure 1: it would have been interesting to see here a multiple sequence alignment between botryllin and previously characterized styelins.

Done. This helped to put in evidence the similarities between botryllin and styelins

Figure 2: the use of letters to indicate both the figure panels and the statistical significance may be confusing. I would suggest to replace the latter with different symbols.

We replaced the letters indicating the statistical significance with Greek letters.

L211: since the genome of the target species is available, were the authors able to rule out the possibility that other paralogous genes were present? This might be relevant to check that ISH signals were specific to this particular transcript.

As explained in the text, in the available transcriptomes and genome of B. schlosseri, we identified three similar sequences from only one transcriptome. They differ in a few nucleotides both at the CDS and UTRs. The high sequence variability found in B. schlosseri, together with a still highly fragmented genome does not allowed us to ascertain whether the above sequences are the result of a recent gene duplication or, more likely, are attributable to isoforms and not to different paralogous genes. Future studies will be aimed at the resolution of this problem.

L284: while the use of a panel of microorganisms of common laboratory use is certainly useful to preliminarily investigate the antimicrobial activity of a novel AMP, I would suggest the authors to remark the fact that botryllin might have a broader spectrum of activity, perhaps showing even stronger killing activity, against a plethora of other microorganisms found in the marine environment that were not tested in the present study (and could not be tested due to the facts that the overwhelming majority of marine bacteria cannot be presently cultured in laboratory conditions).

Thanks for the suggestion. It was added to the discussion.

L304: this is a reasonable consideration, even though the authors cannot exclude that other biological stimuli (e.g. the exposure to other microbes commonly found in the marine environment) can trigger botryllin expression. On the other hand, the massive depletion of botryllin from MC granules would suggest a regulation mode that could involve apparent over-expression several hours after the exposure, in order to replace the AMP storage lost with degranulation.

MC and neighbouring cells die upon degranulation as they release cytotoxic material (the enzynme PO and its polyphenol substrata) that induce oxidative stress.

Reviewer 2 Report

This study described and characterized a novel antimicrobial peptide ‘botryllin’ from Botryllus schlosser Furthermore, the authors also clarified the localization and illustrated the bactericidal process in the tunicate.  This study contribute for better understanding of the innate immune system of the colonial ascidians as well as pharmaceutical applications. I recommend the publication of this article in Marine Drugs.

Minor comments

Line 16: Does “nonself” mean allogeneic tissue/cells? This is uncertain whether it is speculation or observation.

Line 58: Check whether this C. intestinal are really C. intestinqlis snsu strict or C. robusta based on the sampling sites of the organisms.

Line 43-64: Consider to add patellamide and or so-called cyanobactines synthesized by cyanobacterial symbionts. 

 Schmidt et al. 2012. Origin and Variation of Tunicate Secondary Metabolites. Journal of Natural Products 75:295–304. DOI: 10.1021/np200665k.

And also patellazole synthesized by endosymbionts in hemocytes.

Kwan et al. 2012. Genome streamlining and chemical defense in a coral reef symbiosis. PNAS 109:20655–20660.

Line 98. ‘Drosophila’ should be italic.

Line 109—: Some species names should be italic.

Fig 2: Consider to add symbols such as ‘a > b’,   ‘a > b > c’ or so.

Line 316–319. Possible involvement of botrylin in the colonial allo-rejection is very interesting discussion for me and wish to know potential cytotoxic activity of bftrylin against ascidian hemocytes. If the authors have any data on the cytotoxity of botrylin on isogenic and/or allogeneic hemocytes, consider to add them.

Author Response

We thank the anonymous reviewer 2 for his/her comments that helped us to improve the quality of the manuscript. Here below our replies to their comments

Line 16: Does “nonself” mean allogeneic tissue/cells? This is uncertain whether it is speculation or observation.

We changed the term “nonself” with “foreign cells”

Line 58: Check whether this C. intestinal are really C. intestinqlis snsu strict or C. robusta based on the sampling sites of the organisms.

Done. Both the species were used. This was specified in the text.

Line 43-64: Consider to add patellamide and or so-called cyanobactines synthesized by cyanobacterial symbionts. 

Schmidt et al. 2012. Origin and Variation of Tunicate Secondary Metabolites. Journal of Natural Products 75:295–304. DOI: 10.1021/np200665k.

And also patellazole synthesized by endosymbionts in hemocytes.

Kwan et al. 2012. Genome streamlining and chemical defense in a coral reef symbiosis. PNAS 109:20655–20660.

Done. The suggested  papers were cited in the text and added in the reference list

Line 98. ‘Drosophila’ should be italic.

Done

Line 109—: Some species names should be italic.

We checked the text carefully. It should be OK now.

Fig 2: Consider to add symbols such as ‘a > b’,   ‘a > b > c’ or so.

We replaced the letters with Greek letters, according to the suggestins of referee 1.

Line 316–319. Possible involvement of botrylin in the colonial allo-rejection is very interesting discussion for me and wish to know potential cytotoxic activity of bftrylin against ascidian hemocytes. If the authors have any data on the cytotoxity of botrylin on isogenic and/or allogeneic hemocytes, consider to add them.

We do not have such data yet. Their acquisition is among the goals of future research.

Reviewer 3 Report

In this work, the authors identified a transcript for a novel styelin-like antimicrobial peptide--botryllin. The gene is constitutively transcribed by circulating cytotoxic morula cells (MCs). The synthetic peptide, obtained from in silico translation of the transcript, shows robust killing activity to bacterial and unicellular yeast cells causing breakages of both the plasma membrane and the cell wall. The obtained results suggest that the botryllin gene carries the information for the synthesis of an AMP involved in the protection of B. schlosseri from invading foreign cells. The work is important and interesting. I suggest that this manuscript is suitable for publishing after minor revisions.

 In the manuscript, all the ml should be change to mL.

The antimicrobial activity of botryllin was also assayed at various physico-chemical conditions such as temperature, pH and ionic strength to determine the optimum and the resistance/decrease capability of the peptide. Please explain the basis for the selection of test conditions and explain the reasons in combination with the experimental results.

Briefly introduce the prospect of botryllin as a drug application. How about its pharmaceutical properties?

Author Response

We thank the anonymous reviewer 3 for his/her comments that helped us to improve the quality of the manuscript. Here below our replies to their comments

In the manuscript, all the “ml” should be change to “mL”.

Done

The antimicrobial activity of botryllin was also assayed at various physico-chemical conditions such as temperature, pH and ionic strength to determine the optimum and the resistance/decrease capability of the peptide. Please explain the basis for the selection of test conditions and explain the reasons in combination with the experimental results.

We added a sentence in the discussion to explain this point

Briefly introduce the prospect of botryllin as a drug application. How about its pharmaceutical properties?

We added a sentence in the discussion to explain this point